# MUSIC SOURCE SEPARATION IN THE WAVEFORM DOMAIN

## ABSTRACT

Source separation for music is the task of isolating contributions, or *stems*, from different instruments recorded individually and arranged together to form a song. Such components include voice, bass, drums and any other accompaniments. Contrarily to many audio synthesis tasks where the best performances are achieved by models that directly generate the waveform, the state-of-the-art in source separation for music is to compute masks on the magnitude spectrum. In this paper, we first show that an adaptation of Conv-Tasnet (Luo & Mesgarani, 2019), a waveform-to-waveform model for source separation for speech, significantly beats the state-of-the-art on the MusDB dataset, the standard benchmark of multi-instrument source separation. Second, we observe that Conv-Tasnet follows a masking approach on the input signal, which has the potential drawback of removing parts of the relevant source without the capacity to reconstruct it. We propose Demucs, a new waveform-to-waveform model, which has an architecture closer to models for audio generation with more capacity on the decoder. Experiments on the MusDB dataset show that Demucs beats previously reported results in terms of signal to distortion ratio (SDR), but lower than Conv-Tasnet. Human evaluations show that Demucs has significantly higher quality (as assessed by mean opinion score) than Conv-Tasnet, but slightly more contamination from other sources, which explains the difference in SDR. Additional experiments with a larger dataset suggest that the gap in SDR between Demucs and Conv-Tasnet shrinks, showing that our approach is promising.

## 1 INTRODUCTION

Cherry first noticed the "cocktail party effect" (Cherry, 1953): how the human brain is able to separate a single conversation out of a surrounding noise from a room full of people chatting. Bregman later tried to understand how the brain was able to analyse a complex auditory signal and segment it into higher level streams. His framework for auditory scene analysis (Bregman, 1990) spawned its computational counterpart, trying to reproduce or model accomplishments of the brains with algorithmic means (Wang & Brown, 2006), in particular regarding source separation capabilities.

When producing music, recordings of individual instruments called *stems* are arranged together and mastered into the final song. The goal of source separation is to recover those individual stems from the mixed signal. Unlike the cocktail party problem, there is not a single source of interest to differentiate from an unrelated background noise, but instead a wide variety of tones and timbres playing in a coordinated way. In the SiSec Mus evaluation campaign for music separation (Stöter et al., 2018), those individual stems were grouped into 4 broad categories: (1) `drums`, (2) `bass`, (3) `other`, (4) `vocals`. Given a music track which is a mixture of these four sources, also called the mix, the goal is to generate four waveforms that correspond to each of the original sources. We consider here the case of supervised source separation, where the training data contain music tracks (i.e., mixtures), together with the ground truth waveform for each of the sources.

State-of-the-art approaches in music source separation still operate on the spectrograms generated by the short-time Fourier transform (STFT). They produce a mask on the magnitude spectrums for each frame and each source, and the output audio is generated by running an inverse STFT on the masked spectrograms reusing the input mixture phase (Takahashi & Mitsufuji, 2017; Takahashi et al., 2018). Several architectures trained end-to-end to directly synthesize the waveforms have been proposed (Lluís et al., 2018; Jansson et al., 2017), but their performances are far below the state-of-the-art: in

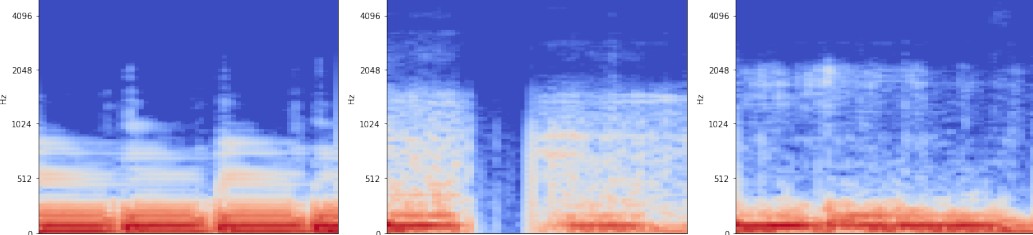

Figure 1: Mel-spectrogram for a 0.8 seconds extract of the `bass` source from the track "Stich Up" of the MusDB test. From left to right: ground truth, Conv-Tasnet estimate and Demucs estimate. We observe that Conv-Tasnet missed one note entirely.

the last SiSec Mus evaluation campaign (Stöter et al., 2018), the best model that directly predicts waveforms achieves an average signal-to-noise ratio (SDR) over all four sources of 3.2, against 5.3 for the best approach that predicts spectrograms masks (also see Table 1 in Section 6). An upper bound on the performance of all methods relying on masking spectrograms is given by the SDR obtained when using a mask computed using the ground truth sources spectrograms, for instance the Ideal Ratio Mask (IRM) or the Ideal Binary Mask (IBM) oracles. For speech source separation, Luo & Mesgarani (2019) proposed Conv-Tasnet, a model that reuses the masking approach of spectrogram methods but learns the masks jointly with a convolutional front-end, operating directly in the waveform domain for both the inputs and outputs. Conv-Tasnet surpasses both the IRM and IBM oracles.

Our first contribution is to adapt the Conv-Tasnet architecture, originally designed for monophonic speech separation and audio sampled at 8 kHz, to the task of sterephonic music source separation for audio sampled at 44.1 kHz. Our experiments show that Conv-Tasnet outperforms all previous methods by a large margin, with an SDR of 5.7, but still under the SDR of the IRM oracle at 8.2 (Stöter et al., 2018). However, while Conv-Tasnet separates with a high accuracy the different sources, we observed artifacts when listening to the generated audio: a constant broadband noise, hollow instruments attacks or even missing parts. They are especially noticeable on the `drums` and `bass` sources and we give one such example on Figure 1. Conv-Tasnet uses an over-complete linear representation on which it applies a mask obtained from a deep convolutional network. Because both the encoder and decoder are linear, the masking operation cannot synthesize new sounds. We conjecture that the overlap of multiples instruments sometimes lead to a loss of information that is not reversible by a masking operation.

To overcome the limitations of Conv-Tasnet, our second contribution is to propose Demucs, a new architecture for music source separation. Similarly to Conv-Tasnet, Demucs is a deep learning model that directly operates on the raw input waveform and generates a waveform for each source. Demucs is inspired by models for music synthesis rather than masking approaches. It is a U-net architecture with a convolutional encoder and a decoder based on wide transposed convolutions with large strides inspired by recent work on music synthesis (Défossez et al., 2018). The other critical features of the approach are a bidirectional LSTM between the encoder and the decoder, increasing the number of channels exponentially with depth, gated linear units as activation function (Dauphin et al., 2017) which also allow for masking, and a new initialization scheme.

We present experiments on the MusDB benchmark, which first show that both Conv-Tasnet and De-mucs achieve performances significantly better than the best methods that operate on the spectrogram, with Conv-Tasnet being better than Demucs in terms of SDR. We also perform human evaluations that compare Conv-Tasnet and our Demucs, which show that Demucs has significantly better perceived quality. The smaller SDR of Demucs is explained by more contamination from other sources. We also conduct an in-depth ablation study of the Demucs architecture to demonstrate the impact of the various design decisions. Finally, we carry out additional experiments by adding 150 songs to the training set. In this experiment, Demucs and TasNet both achieve an SDR of 6.3, suggesting that the gap in terms of SDR between the two models diminishes with more data, making the Demucs approach promising. The 6.3 points of SDR also set a new state-of-the-art, since it improves on the best previous result of 6.0 on the MusDB test set obtained by training with 800 additional songs.

We discuss in more detail the related work in the next Section. We then describe the original Conv-Tasnet model of Luo & Mesgarani (2018) and its adaptation to music source separation. Our Demucs

architecture is detailed in Section 4. We present the experimental protocol in Section 5, and the experimental results compared to the state-of-the-art in Section 6. Finally, we describe the results of the human evaluation and the ablation study.

## 2 RELATED WORK

A first category of methods for supervised music source separation work on time-frequency representations. They predict a power spectrogram for each source and reuse the phase from the input mixture to synthesise individual waveforms. Traditional methods have mostly focused on blind (unsupervised) source separation. Non-negative matrix factorization techniques (Smaragdis et al., 2014) model the power spectrum as a weighted sum of a learnt spectral dictionary, whose elements are grouped into individual sources. Independent component analysis (Hyvärinen et al., 2004) relies on independence assumptions and multiple microphones to separate the sources. Learning a soft/binary mask over power spectrograms has been done using either HMM-based prediction (Roweis, 2001) or segmentation techniques (Bach & Jordan, 2005).

With the development of deep learning, fully supervised methods have gained momentum. Initial work was performed on speech source separation (Grais et al., 2014), followed by works on music using simple fully connected networks over few spectrogram frames (Uhlich et al., 2015), LSTMs (Uhlich et al., 2017), or multi scale convolutional/recurrent networks (Liu & Yang, 2018; Takahashi & Mitsufuji, 2017). Nugraha et al. (2016) showed that Wiener filtering is an efficient post-processing step for spectrogram-based models and it is now used by all top performing models in this category. Those methods have performed the best during the last SiSec 2018 evaluation (Stöter et al., 2018) for source separation on the MusDB (Rafii et al., 2017) dataset. After the evaluation, a reproducible baseline called Open Unmix has been released by Stöter et al. (2019) and matches the top submissions trained only on MusDB. MMDenseLSTM, a model proposed by Takahashi et al. (2018) and trained on 807 unreleased songs currently holds the absolute record of SDR in the SiSec campaign. Both Demucs and Conv-Tasnet obtain significantly higher SDR.

More recently, models operating in the waveform domain have been developed, so far with worse performance than those operating in the spectrogram domain. A convolutional network with a U-Net structure called Wave-U-Net was used first on spectrograms (Jansson et al., 2017) and then adapted to the waveform domain (Stoller et al., 2018). Wave-U-Net was submitted to the SiSec 2018 evaluation campaign with a performance inferior to that of most spectrogram domain models by a large margin. A Wavenet-inspired, although using a regression loss and not auto-regressive, was first used for speech denoising (Rethage et al., 2018) and then adapted to source separation (Lluís et al., 2018). Our model significantly outperforms Wave-U-Net.Given that the Wavenet inspired model performed worse than Wave-U-Net, we did not consider it for comparison.

In the field of monophonic speech source separation, spectrogram masking methods have enjoyed good performance (Kolbæk et al., 2017; Isik et al., 2016). Luo & Mesgarani (2018) developed a waveform domain methods using masking over a learnable front-end obtained from a LSTM that reached the same accuracy. Improvements were obtained by Wang et al. (2018) for spectrogram methods using the unfolding of a few iterations of a phase reconstruction algorithm in the training loss. In the mean time, Luo & Mesgarani (2019) refined their approach, replacing the LSTM with a superposition of dilated convolutions, which improved the SDR and definitely surpassed spectrogram based approaches, including oracles that use the ground truth sources such as the ideal ratio mask (IRM) or the ideal binary mask (IBM). We show in this paper that Conv-Tasnet also outperforms all known methods for music source separation. However it suffers from significantly more artifacts than the Demucs architecture we introduce in this paper, as measured by mean opinion score.

## 3 ADAPTING CONV-TASNET FOR MUSIC SOURCE SEPARATION

We describe in this section the Conv-Tasnet architecture of Luo & Mesgarani (2018) and give the details of how we adapted the architecture to fit the setting of the MusDB dataset.

**Overall framework** Each source $s$ is represented by a waveform $x_s \in \mathbb{R}^{C,T}$ where $C$ is the number of channels (1 for mono, 2 for stereo) and $T$ the number of samples of the waveform. The mixture (i.e., music track) is the sum of all sources $x := \sum_{s=1}^{S} x_s$. We aim at training a model $g$

parameterized by $\theta$, such that $g(x) = (g_s(x;\theta))_{s=1}^S$, where $g_s(x;\theta)$ is the predicted waveform for source $s$ given $x$, that minimizes

$$\min_\theta \sum_{x \in \mathcal{D}} \sum_{s=1}^S L(g_s(x;\theta), x_s) \tag{1}$$

for some dataset $\mathcal{D}$ and reconstruction error $L$. The original Conv-Tasnet was trained using a loss called scale-invariant source-to-noise ratio (SI-SNR), similar to the SDR loss described in Section 5. We instead use a simple L1 loss between the estimated and ground truth sources. We discuss in more details regression losses in the context of our Demucs architecture in Section 4.2.

**The original Conv-Tasnet architecture**   Conv-Tasnet (Luo & Mesgarani, 2018) is composed of a learnt front-end that transforms back and forth between the input monophonic mixture waveform sampled at 8 kHz and a 128 channels over-complete representation sampled at 1 kHz using a convolution as the encoder and a transposed convolution as the decoder, both with a kernel size of 16 and stride of 8. The high dimensional representation is masked through a separation network composed of stacked residual blocks. Each block is composed of a a 1x1 convolution, a PReLU (He et al., 2015b) non linearity, a layer wise normalization over all channels jointly (Ba et al., 2016), a depth-wise separable convolution (Chollet, 2017; Howard et al., 2017) with a kernel size of 3, a stride of 1 and a dilation of $2^{n \bmod N}$, with $n$ the 0-based index of the block and $N$ an hyper-parameter, and another PReLU and normalization. The output of each block participates to the final mask estimation through a skip connection, preceded by a 1x1 convolution. The original Conv-Tasnet counted $3 \times N$ blocks with $N = 8$. The mask is obtained summing the output of all blocks and then applying ReLU. The output of the encoder is multiplied by the mask and before going through the decoder.

**Conv-Tasnet for music source separation**   We adapted their architecture to the task of stereophonic music source separation: the original Conv-Tasnet has a receptive field of 1.5 seconds for audio sampled at 8 kHz, we take $N = 10$ and increased the kernel size (resp. stride) of the encoder/decoder from 16 (resp. 8) to 20 (resp. 10), leading to the same receptive field at 44.1 kHz. We observed better results using $4 \times N$ blocks instead of $3 \times N$ and 256 channels for the encoder/decoder instead of 128. With those changes, Conv-Tasnet obtained state-of-the-art performance on the MusDB dataset, surpassing all known spectrogram based methods by a large margin as shown in Section 6.

**Separating entire songs**   The original Conv-Tasnet model was designed for short sentences of a few seconds at most. When evaluating it on an entire track, we obtained the best performance by first splitting the input track into chunks of 8 seconds each. We believe this is because of the global layer normalization. During training, only small audio extracts are given, so that a quiet part or a loud part would be scaled back to an average volume. However, when using entire songs as input, it will most likely contain both quiet and loud parts. The normalization will not map both to the same volume, leading to a difference between training and evaluation. We did not observe any side effects when going from one chunk to the next, so we did not look into fancier overlap-add methods.

## 4  THE DEMUCS ARCHITECTURE

The architecture we propose, which we call Demucs, is described in the next few subsections, and the reconstruction loss is discussed in Section 4.2. Demucs takes a stereo mixture as input and outputs a stereo estimate for each source ($C = 2$). It is an encoder/decoder architecture composed of a convolutional encoder, a bidirectional LSTM, and a convolutional decoder, with the encoder and decoder linked with skip U-Net connections. Similarly to other work in generation in both image (Karras et al., 2018; 2017) and sound (Défossez et al., 2018), we do not use batch normalization (Ioffe & Szegedy, 2015) as our early experiments showed that it was detrimental to the model performance. The overall architecture is depicted in Figure 2a.

### 4.1  CONVOLUTIONAL AUTO-ENCODER

**Encoder**   The encoder is composed of $L := 6$ stacked convolutional blocks numbered from 1 to $L$. Each block $i$ is composed of a convolution with kernel size $K := 8$, stride $S := 4$, $C_{i-1}$ input channels, $C_i$ output channels and ReLU activation, followed by a convolution with kernel size 1, $2C_i$

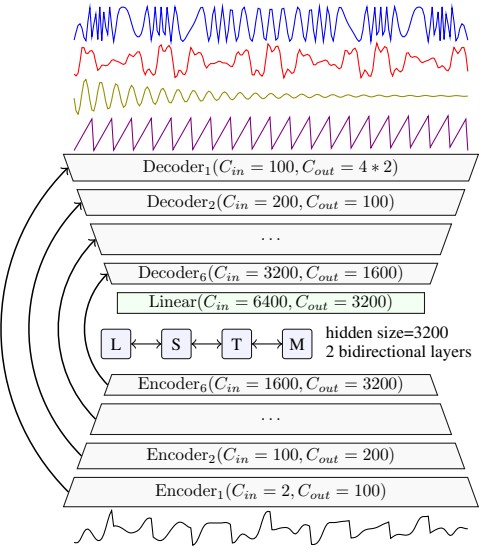

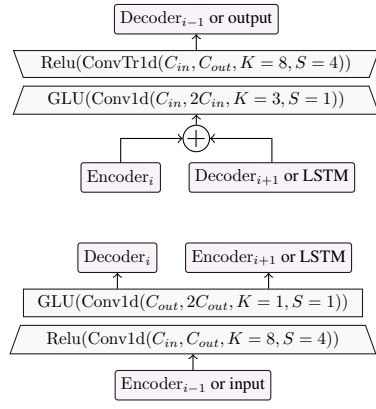

(a) Demucs architecture with the mixture waveform as input and the four sources estimates as output. Arrows represents U-Net connections.

(b) Detailed view of the layers $\text{Decoder}_i$ on the top and $\text{Encoder}_i$ on the bottom. Arrows represent connections to other parts of the model. For convolutions, $C_in$ (resp $C_out$) is the number of input channels (resp output), $K$ the kernel size and $S$ the stride.

Figure 2: Demucs complete architecture on the left, with detailed representation of the encoder and decoder layers on the right.

output channels and gated linear units (GLU) as activation function (Dauphin et al., 2017). Since GLUs halve the number of channels, the final output of block $i$ has $C_i$ output channels. A block is described in Figure 2b. Convolutions with kernel width 1 increase the depth and expressivity of the model at low computational cost. As we show in our ablation study 6.2, the usage of GLU activations after these convolutions significantly boost performance. The number of channels in the input mixture is $C_0 = C = 2$, while we use $C_1 := 100$ as the number of output channels for the first encoder block. The number of channels is then doubled at each subsequent block, i.e., $C_i := 2C_{i-1}$ for $i = 2..L$, so the final number of channels is $C_L = 3200$. We then use a bidirectional LSTM with 2 layers and a hidden size $C_L$. The LSTM outputs $2C_L$ channels per time position. We use a linear layer to take that number down to $C_L$.

**Decoder** The decoder is mostly the inverse of the encoder. It is composed of $L$ blocks numbered in reverse order from $L$ to $1$. The $i$-th blocks starts with a convolution with stride 1 and kernel width 3 to provide context about adjacent time steps, input/output channels $C_i$ and a ReLU activation. Finally, we use a transposed convolution with kernel width 8 and stride 4, $C_{i-1}$ outputs and ReLU activation. The $S$ sources are synthesized at the final layer only, after all decoder blocks. The final layer is linear with $S \cdot C_0$ output channels, one for each source (4 stereo channels in our case), without any additional activation function. Each of these channels directly generate the corresponding waveform.

**U-network structure** Similarly to Wave-U-Net (Jansson et al., 2017), there are skip connections between the encoder and decoder blocks with the same index, as originally proposed in U-networks (Ronneberger et al., 2015). While the main motivation comes from empirical performances, an advantage of the skip connections is to give a direct access to the original signal, and in particular allows to directly transfers the phase of the input signal to the output, as discussed in Section 4.2. Unlike Wave-U-Net, we use transposed convolutions rather than linear interpolation followed by a convolution with a stride of 1. For the same increase in the receptive field, transposed convolutions require 4 times less operations and memory. This limits the overall number of channels that can be used before running out of memory. As we observed that a large number of channels was key to obtaining good results, we favored the use of transposed convolutions, as explained in Section 6.

**Motivation: synthesis vs masking** The approach we follow uses the U-Network architecture (Ronneberger et al., 2015; Stoller et al., 2018; Jansson et al., 2017), and builds on transposed convolutions with large number of channels and large strides (4) inspired by the approach to the synthesis of music notes of Défossez et al. (2018). The U-Net skip connections and the gating

performed by GLUs imply that this architecture is expressive enough to represent masks on a learnt representation of the input signal, in a similar fashion to Conv-Tasnet. The Demucs approach is then more expressive than Conv-Tasnet, and its main advantages are the multi-scale representations of the input and the non-linear transformations to and from the waveform domain.

## 4.2 LOSS FUNCTION

For the reconstruction loss $L(g_s(x; \theta), x_s)$ in equation 1, we either use the average mean square error or average absolute error between waveforms: for a waveform $x_s$ containing $T$ samples and corresponding to source $s$, a predicted waveform $\hat{x}_s$ and denoting with a subscript $t$ the $t$-th sample of a waveform, we use one of $L_1$ or $L_2$:

$$L_1(\hat{x}_s, x_s) = \frac{1}{T} \sum_{t=1}^{T} |\hat{x}_{s,t} - x_{s,t}| \qquad L_2(\hat{x}_s, x_s) = \frac{1}{T} \sum_{t=1}^{T} (\hat{x}_{s,t} - x_{s,t})^2. \qquad (2)$$

In generative models for audio, direct reconstruction losses on waveforms can pose difficulties because they are sensitive to the initial phases of the signals: two signals whose only difference is a shift in the initial phase are perceptually the same, but can have arbitrarily high $L_1$ or $L_2$ losses. It can be a problem in pure generation tasks because the initial phase of the signal is unknown, and losses on power/magnitude spectrograms are alternative that do not suffer from this lack of specification of the output. Approaches that follow this line either generate spectrograms (e.g., Wang et al., 2017), or use a loss that compares power spectrograms of target/generated waveforms (Défossez et al., 2018).

The problem of invariance to a shift of phase is not as severe in source separation as it is in unconditional generation, because the model has access to the original phase of the signal. The pase can easily be recovered from the skip connections in U-net-style architectures for separation, and is directly used as input of the inverse STFT for methods that generate masks on power spectrograms. As such, losses such as $L1/L2$ are totally valid for source separation. Early experiments with an additional term including the loss of Défossez et al. (2018) did not suggest that it boosts performance, so we did not pursue this direction any further. Most our experiments use $L1$ loss, and the ablation study presented in Section 6.2 suggests that there is no significant difference between $L1$ and $L2$.

## 4.3 WEIGHT RESCALING AT INITIALIZATION

The initialization of deep neural networks is known to have a critical impact on the overall performances (Glorot & Bengio, 2010; He et al., 2015a), up to the point that Zhang et al. (2019) showed that with a different initialization called fixup, very deep residual networks and transformers can be trained without batch normalization. While Fixup is not designed for U-Net-style skip connections, we observed that the following different initialisation scheme had great positive impact on performances compared to the standard initialization of He et al. (2015a) used in U-Networks.

Considering the so-called Kaiming initialization (He et al., 2015a) as a baseline, let us look at a single convolution layer for which we denote $w$ the weights after the first initialization. We take $\alpha := \text{std}(w)/a$, where $a$ is a reference scale, and replace $w$ by $w' = w/\sqrt{\alpha}$. Since the original weights have element-wise order of magnitude $(KC_{\text{in}})^{-1/2}$ where $K$ is the kernel width and $C_{\text{in}}$ the number of output channels, it means that our initialization scheme produces weights of order of magnitude $(KC_{\text{in}})^{-1/4}$, together with a non-trivial scale. Based a search over the values [0.01, 0.05, 0.1], we select $a = 0.1$ for all the regular and transposed convolutions, see Section 6 for more details. We experimentally observed that on a randomly initialized model applied to an audio extract, it kept the standard deviation of the features along the layers of the same order of magnitude. Without initial rescaling, the output the last layer has a magnitude 20 times smaller than the first.

## 4.4 RANDOMIZED EQUIVARIANT STABILIZATION

A perfect source separation model is time equivariant, i.e. shifting the input mixture by X samples will shift the output Y by the exact same amount. Thanks to its dilated convolutions with a stride of 1, the mask predictor of Conv-Tasnet is naturally time equivariant and even if the encoder/decoder is not strictly equivariant, Conv-Tasnet still verifies this property experimentally (Luo & Mesgarani, 2019). Spectrogram based method will also verify approximately this property. Shifting the input

by a small amount will only reflect in the phase of the spectrogram. As the mask is computed only from the magnitude, and the input mixture phase is reused, the output will naturally be shifted by the same amount. On the other hand, we noticed that our architecture did not naturally satisfy this property. We propose a simple workaround called randomized equivariant stabilization, where we sample $S$ random shifts of an input mixture $x$ and average the predictions of our model for each, after having applied the opposite shift. This technique does not require changing the training procedure or network architecture. Using $S = 10$, we obtained a 0.3 SDR gain, see Section 6.2 for more details. It does make evaluation of the model $S$ times slower, however, on a V100 GPU, separating 1 minute of audio at 44.1 kHz with Demucs takes only 0.8 second. With this technique, separation of 1 minute takes 8 seconds which is still more than 7 times faster than real time.

## 5 EXPERIMENTAL SETUP

### 5.1 EVALUATION FRAMEWORK

**MusDB and additional data**    We use the MusDB dataset (Rafii et al., 2017) , which is composed of 150 songs with full supervision in stereo and sampled at 44100Hz. For each song, we have the exact waveform of the `drums`, `bass`, `other` and `vocals` parts, i.e. each of the sources. The actual song, the mixture, is the sum of those four parts. The first 84 songs form the *train set*, the next 16 songs form the *valid set* (the exact split is defined in the `musdb` python package) while the remaining 50 are kept for the *test set*. We collected raw stems for 150 tracks, i.e., individual instrument recordings used in music production software to make a song. We manually assigned each instrument to one of the sources in MusDB. We call this extra supervised data the *stem set*. We also report the performances of Tasnet and Demucs trained using these 150 songs in addition to the 84 from MusDB, to anaylze the effect of adding more training data.

**Source separation metrics**    Measurements of the performance of source separation models was developed by Vincent et al. for blind source separation (Vincent et al., 2006) and reused for supervised source separation in the SiSec Mus evaluation campaign (Stöter et al., 2018). Similarly to previous work (Stoller et al., 2018; Takahashi & Mitsufuji, 2017; Takahashi et al., 2018), we focus on the SDR (Signal to Distortion Ratio) which measures the log ratio between the volume of the estimated source projection onto the ground truth, and the volume of what is left out of this projection, typically contamination by other sources or artifacts. Other metrics can be defined (SIR and SAR) and we present them in the supplementary material. We used the python package `museval` which provide a reference implementation for the SiSec Mus 2018 evaluation campaign. As done in the SiSec Mus competition, we report the median over all tracks of the median of the metric over each track computed using the `museval` package.

### 5.2 BASELINES

As baselines, we selected Open Unmix (Stöter et al., 2019), a 3-layer BiLSTM model with encoding and decoding fully connected layers on spectrogram frames. It was released by the organizers of the SiSec 2018 to act as a strong reproducible baseline and matches the performances of the best candidates trained only on MusDB. We also selected MMDenseLSTM (Takahashi et al., 2018), a multi-band dense net with LSTMs at different scales of the encoder and decoder. This model was submitted as TAK2 and trained with 804 extra labeled songs[1]. Both MMDenseLSTM and Open Unmix use Wiener filtering (Nugraha et al., 2016) as a last post processing step. The only waveform based method submitted to the evaluation campaign is Wave-U-Net (Stoller et al., 2018) with the identifier STL2. Metrics were downloaded from the SiSec submission repository. for Wave-U-Net and MMDenseLSTM. For Open Unmix they were provided by their authors[2]. We also provide the metrics for the Ideal Ratio Mask oracle (IRM), which computes the best possible mask using the ground truth sources and is the topline of spectrogram based method (Stöter et al., 2018).

---

[1]Source: `https://sisec18.unmix.app/#/methods/TAK2`
[2]`https://zenodo.org/record/3370486`

Table 1: Comparison of Conv-Tasnet and Demucs to state-of-the-art models that operate on the waveform (Wave-U-Net) and on spectrograms (Open-Unmix without extra data, MMDenseLSTM with extra data) as well as the IRM oracle on the MusDB test set. The *Extra?* indicates the number of extra training songs used. We report the median over all tracks of the median SDR over each track, as done in the SiSec Mus evaluation campaign (Stöter et al., 2018). The `All` column reports the average over all sources. Demucs metrics are averaged over 5 runs, the confidence interval is the standard deviation over $\sqrt{5}$. In bold are the values that are statistically state-of-the-art either with or without extra training data.

| | | | Test SDR in dB | | | | |
|---|---|---|---|---|---|---|---|
| **Architecture** | **Wav?** | **Extra?** | `All` | `Drums` | `Bass` | `Other` | `Vocals` |
| IRM oracle | ✗ | N/A | 8.22 | 8.45 | 7.12 | 7.85 | 9.43 |
| Open-Unmix | ✗ | ✗ | 5.33 | 5.73 | 5.23 | 4.02 | 6.32 |
| Wave-U-Net | ✓ | ✗ | 3.23 | 4.22 | 3.21 | 2.25 | 3.25 |
| Demucs | ✓ | ✗ | 5.58 $\pm$.03 | **6.08** $\pm$.06 | **5.83** $\pm$.07 | 4.12 $\pm$.04 | 6.29 $\pm$.07 |
| Conv-Tasnet | ✓ | ✗ | **5.73** $\pm$.03 | **6.08** $\pm$.06 | 5.66 $\pm$.16 | **4.37** $\pm$.02 | **6.81** $\pm$.04 |
| Demucs | ✓ | 150 | **6.33** $\pm$.02 | **7.08** $\pm$.07 | 6.70 $\pm$.06 | 4.47 $\pm$.03 | 7.05 $\pm$.04 |
| Conv-Tasnet | ✓ | 150 | **6.32** $\pm$.04 | **7.11** $\pm$.13 | 7.00 $\pm$.05 | 4.44 $\pm$.03 | 6.74 $\pm$.06 |
| MMDenseLSTM | ✗ | 804 | 6.04 | 6.81 | 5.40 | **4.80** | **7.16** |

## 5.3 TRAINING PROCEDURE

**Epoch definition and augmentation**  We define one epoch over the dataset as a pass over all 11-second extracts with a stride of 1 second. We use a random audio shift between 0 and 1 second and keep 10 seconds of audio from there as a training example. We perform the following data augmentation (Uhlich et al., 2017), also used by Open Unmix and MMDenseLSTM: shuffling sources within one batch to generate one new mix, randomly swapping channels. We additionally multiply each source by $\pm 1$ (Nachmani & Wolf, 2019). All Demucs models were trained over 240 epochs. Conv-Tasnet was trained for 360 epochs when trained only on MusDB and 240 when trained with extra data and using only 2-seconds audio extracts.

**Training setup and hyper-parameters**  All models are trained with 16 V100 GPUs with 32GB of RAM. We use a batch size of 64, the Adam (Kingma & Ba, 2015) optimizer with a learning rate was chosen among [3e-4, 5e-4] and the initial number of channels was chosen in [64, 80, 100] based on the L1 loss on the validation set. We obtained best performance with a learning rate of $3e-4$ and 100 channels. We then tried 3 different values for the initial weight rescaling reference level described in Section 4.3, [0.01, 0.05, 0.1] and selected 0.1. We computed confidence intervals using 5 random seeds in Table 1. For the ablation study on Table 4, we provide metrics for a single run.

## 6 EXPERIMENTAL RESULTS

In this section, we first provide experimental results on the MusDB dataset for Conv-Tasnet and Demucs compared with state-of-the-art baselines. We then dive into the ablation study of Demucs.

## 6.1 COMPARISON WITH BASELINES

We provide a comparison the state-of-the-art baselines on Table 1. The models on the top half were trained without any extra data while the lower half used unreleased training songs. As no previous work included confidence intervals, we considered the single metric provided by for the baselines as the exact estimate of their mean performance.

**Quality of the separation**  We first observe that Demucs and Conv-Tasnet outperforms all previous methods for music source separation. Conv-Tasnet has significantly higher SDR with 5.73, improving by 0.4 over Open-Unmix. Our proposed Demucs architecture has worse overall performance but

Table 2: Mean Opinion Scores (MOS) evaluating the quality and absence of artifacts of the separated audio. 38 people rated 20 samples each, randomly sample from one of the 3 models or the ground truth. There is one sample per track in the MusDB test set and each is 8 seconds long. Ratings of 5 means that the quality is perfect (no artifacts).

| Architecture | Quality Mean Opinion Score | | | | |
|---|---|---|---|---|---|
| | All | Drums | Bass | Other | Vocals |
| Ground truth | 4.46 ±.07 | 4.56 ±.13 | 4.25 ±.15 | 4.45 ±.13 | 4.64 ±.13 |
| Open-Unmix | 3.03 ±.09 | 3.10 ±.17 | 2.93 ±.20 | 3.09 ±0.16 | 3.00 ±.17 |
| Demucs | 3.22 ±.09 | 3.77 ±.15 | 3.26 ±.18 | 3.32 ±.15 | 2.55 ±.20 |
| Conv-Tasnet | 2.85 ±.08 | 3.39 ±.14 | 2.29 ±.15 | 3.18 ±.14 | 2.45 ±.16 |

Table 3: Mean Opinion Scores (MOS) evaluating contamination by other sources. 38 people rated 20 samples each, randomly sampled from one of the 3 models or the ground truth. There is one sample per track in the MusDB test set and each is 8 seconds long. Ratings of 5 means no contamination by other sources.

| Architecture | Contamination Mean Opinion Score | | | | |
|---|---|---|---|---|---|
| | All | Drums | Bass | Other | Vocals |
| Ground truth | 4.59 ±.07 | 4.44 ±.18 | 4.69 ±.09 | 4.46 ±.13 | 4.81 ±.11 |
| Open-Unmix | 3.27 ±.11 | 3.02 ±.19 | 4.00 ±.20 | 3.11 ±.21 | 2.91 ±.20 |
| Demucs | 3.30 ±.10 | 3.08 ±.21 | 3.93 ±.18 | 3.15 ±.19 | 3.02 ±.20 |
| Conv-Tasnet | 3.42 ±.09 | 3.37 ±.17 | 3.73 ±.18 | 3.46 ±.17 | 3.10 ±.17 |

matches Conv-Tasnet for the `drums` source and surpasses it for the `bass`. When training on 150 extra songs, the two methods have the same overall performance of 6.3 SDR, beating MMDenseLSTM by nearly 0.3 SDR, despite MMDenseLSTM being tained on 804 extra songs. Unlike for speech separation (Luo & Mesgarani, 2019), all methods are still far below the IRM oracle, leaving room for future improvements. We provide results for the other metrics (SIR and SAR) as well as box plots with quantiles over the test set tracks in Appendix B. Audio samples for Demucs, Conv-Tasnet and all baselines are provided in the ICLR link code, with more details given in Appendix A.

**Human evaluations** We noticed strong artifacts on the audio separated by Conv-Tasnet, especially for the `drums` and `bass` sources: static noise between 1 and 2 kHz, hollow instrument attacks or missing notes as illustrated on Figure 1. In order to confirm this observation, we organized a mean opinion score survey. We separated 8 seconds extracts from all of the 50 test set tracks for Conv-Tasnet, Demucs and Open-Unmix. We asked 38 participants to rate 20 samples each, randomly taken from one of the 3 models or the ground truth. For each one, they were required to provide 2 ratings on a scale of 1 to 5. The first one evaluated the quality and absence of artifacts (1: many artifacts and distortion, content is hardly recognizable, 5: perfect quality, no artifacts) and the second one evaluated contamination by other sources (1: contamination if frequent and loud, 5: no contamination). We show the results on Tables 2 and 3. We confirmed that the presence of artifacts in the output of Conv-Tasnet degrades the user experience, with a rating of 2.85±.08 against 3.22 ± .09 for Demucs. On the other hand, Conv-Tasnet samples had less contamination by other sources than Open-Unmix or Demucs, although by a small margin, with a rating of 3.42 ± .09 against 3.30 ± .10 for Demucs and 3.27 ± .11 for Open-Unmix.

**Training speed** We measured the time taken to process a single batch of size 16 with 10 seconds of audio at 44.1kHz (the original Wave-U-Net being only trained on 22 kHz audio, we double the time for fairness), ignoring data loading and using `torch.cuda.synchronize` to wait on all kernels to be completed. MMDenseLSTM does not provide a reference implementation. Wave-U-Net takes 1.2 seconds per batch, Open Unmix 0.2 seconds per batch and Demucs 1.6 seconds per batch.

Table 4: Ablation study for the novel elements in our architecture described in Section 4. We use only the train set from MusDB and report best L1 loss over the valid set throughout training as well the SDR on the test set for the epoch that achieved this loss.

| Difference | Valid set L1 loss | Test set SDR |
|---|---|---|
| no initial weight rescaling | 0.172 | 4.94 |
| no BiLSTM | 0.175 | 5.12 |
| ReLU instead of GLU | 0.177 | 5.19 |
| no 1x1 convolutions in encoder | 0.176 | 5.30 |
| no randomized equivariant stabilization | N/A | 5.34 |
| kernel size of 1 in decoder convolutions | 0.166 | 5.51 |
| MSE loss | N/A | 5.55 |
| Reference | 0.164 | 5.58 |

Conv-Tasnet cannot be trained with such a large sample size, however a single iteration over 2 seconds of audio with a batch size of 4 takes 0.7 seconds.

## 6.2 ABLATION STUDY FOR DEMUCS

We provide an ablation study of the main design decisions for Demucs in Table 4. Given the cost of training a single model, we did not compute confidence intervals for each variation. Yet, any difference inferior to .06, which is the standard deviation observed over 5 repetitions of the Reference model, could be attributed to noise.

We observe a small but not significant improvement when using the L1 loss instead of the MSE loss. Adding a BiLSTM and using the initial weight rescaling described in Section 4.3 provides significant gain, with an extra 0.48 SDR for the first and 0.64 for the second. We observe that using randomized equivariant stabilization as described in Section 4 gives a gain of almost 0.3 SDR. We did not report the validation loss as we only use the stabilization when computing the SDR over the test set. We applied the randomized stabilization to Open-Unmix and Conv-Tasnet with no gain, since, as explained in Section 4.4, both are naturally equivariant with respect to initial time shifts.

We introduced extra convolutions in the encoder and decoder, as described in Sections 4.1. The two proved useful, improving the expressivity of the model, especially when combined with GLU activation. Using a kernel size of 3 instead of 1 in the decoder further improves performance. We conjecture that the context from adjacent time steps helps the output of the transposed convolutions to be consistent through time and reduces potential artifacts arising from using a stride of 4.

## CONCLUSION

We showed that Conv-Tasnet, a state-of-the-art architecture for speech source separation that predicts masks on a learnt front-end over the waveform domain, achieves state-of-the-art performance for music source separation, improving over all previous spectrogram or waveform domain methods by 0.4 SDR. While Conv-Tasnet has excellent performance to separate sources, it suffers from noticeable artifacts as confirmed by human evaluations. We developed an alternative approach, Demucs, that combines the ability to mask over a learnt representation with stronger decoder capacity that allows for audio synthesis. We conjecture that this can be useful when information is lost in the mix of instruments and cannot simply be recovered by masking. We show that our approach produces audio of significantly higher quality as measures by mean opinion scores and matches the SDR of Conv-Tasnet when trained with 150 extra tracks. We believe those results make it a promising alternative to methods based on masking only.

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

# APPENDIX

## A  AUDIO SAMPLES

We provide audio samples taken from the test set of MusDB. They are available through the ICLR code sharing url[3] along with all the source code to reproduce our experiments. The audio files for the Wave-U-Net and MMDenseLSTM have been obtained from the SiSec Mus 2018 evaluation campaign results website[4]. For Open Unmix, we generated them from the pretrained UMX model using the reference PyTorch implementation[5]. We recommend listening to the audio samples with headphones, while being careful with the volume. An HTML page `index.html` is provided for easier comparison. The following folders are provided:

- Reference: ground truth,
- Open Unmix,
- WaveUNet,
- Demucs: trained only on MusDB,
- DemucsExtra: trained on MusDB and an extra 150 songs,
- ConvTasnet: trained only on MusDB,
- ConvTasnetExtra: trained on MusDB and an extra 150 songs,
- MMDenseNetLSTM, trained on MusDB and an extra 804 songs.

## B  RESULTS FOR ALL METRICS WITH BOX PLOTS

Reusing the notations from  Vincent et al. (2006), let us take a source $j \in 1, 2, 3, 4$ and introduce $P_{s_j}$ (resp $P_{\mathbf{s}}$) the orthogonal projection on $s_j$ (resp on $\mathrm{Span}(s_1, \ldots, s_4)$). We then take with $\hat{s}_j$ the estimate of source $s_j$

$$s_{\text{target}} := P_{s_j}(\hat{s}_j) \qquad e_{\text{interf}} := P_{\mathbf{s}}(\hat{s}_j) - P_{s_j}(\hat{s}_j) \qquad e_{\text{artif}} := \hat{s}_j - P_{\mathbf{s}}(\hat{s}_j)$$

We can now define various signal to noise ratio, expressed in decibels (dB): the source to distortion ratio

$$\text{SDR} := 10 \log_{10} \frac{\|s_{\text{target}}\|^2}{\|e_{\text{interf}} + e_{\text{artif}}\|^2},$$

the source to interference ratio

$$\text{SIR} := 10 \log_{10} \frac{\|s_{\text{target}}\|^2}{\|e_{\text{interf}}\|^2}$$

and the sources to artifacts ratio

$$\text{SAR} := 10 \log_{10} \frac{\|s_{\text{target}} + e_{\text{interf}}\|^2}{\|e_{\text{artif}}\|^2}.$$

As explained in the main paper, extra invariants are added when using the `museval` package. We refer the reader to Vincent et al. (2006) for more details. We provide box plots for each metric and each target on Figure 3, generated using the notebook provided specifically by the organizers of the SiSec Mus evaluation campaign[6]. Hereafter, we provide the equivalent of Table 1 in the main paper for both SIR and SAR.

---

[3]`https://www.dropbox.com/sh/o0gps94s120v7l4/AABS5vDfuuRjgY_zDjdSm_Fsa?dl=1`
[4]`https://sisec18.unmix.app`
[5]`https://github.com/sigsep/open-unmix-pytorch.`
[6]`https://github.com/sigsep/sigsep-mus-2018-analysis`

| Architecture | Wav? | Extra? | Test SIR in dB | | | | |
|---|---|---|---|---|---|---|---|
| | | | All | Drums | Bass | Other | Vocals |
| IRM oracle | ✗ | N/A | 15.53 | 15.61 | 12.88 | 12.84 | 20.78 |
| Open-Unmix | ✗ | ✗ | 10.49 | 11.12 | 10.93 | 6.59 | 13.33 |
| Wave-U-Net | ✓ | ✗ | 6.26 | 8.83 | 5.78 | 2.37 | 8.06 |
| Demucs | ✓ | ✗ | 10.39 $\pm$.07 | 11.81 $\pm$.27 | 10.55 $\pm$.20 | 5.90 $\pm$.04 | 13.31 $\pm$.21 |
| Conv-Tasnet | ✓ | ✗ | 11.47 $\pm$.09 | 12.31 $\pm$.09 | 11.52 $\pm$.15 | 7.76 $\pm$.07 | 14.30 $\pm$.32 |
| Demucs | ✓ | 150 | 11.95 $\pm$.09 | 13.74 $\pm$.25 | 13.03 $\pm$.22 | 7.11 $\pm$.10 | 13.94 $\pm$.10 |
| Conv-Tasnet | ✓ | 150 | 12.24 $\pm$.09 | 13.66 $\pm$.14 | 13.18 $\pm$.13 | 8.40 $\pm$.08 | 13.70 $\pm$.22 |
| MMDenseLSTM | ✗ | 804 | 12.24 | 11.94 | 11.59 | 8.94 | 16.48 |

| Architecture | Wav? | Extra? | Test SAR in dB | | | | |
|---|---|---|---|---|---|---|---|
| | | | All | Drums | Bass | Other | Vocals |
| IRM oracle | ✗ | N/A | 8.31 | 8.40 | 7.40 | 7.93 | 9.51 |
| Open-Unmix | ✗ | ✗ | 5.90 | 6.02 | 6.34 | 4.74 | 6.52 |
| Wave-U-Net | ✓ | ✗ | 4.49 | 5.29 | 4.64 | 3.99 | 4.05 |
| Demucs | ✓ | ✗ | 6.08 $\pm$.01 | 6.18 $\pm$.03 | 6.41 $\pm$.05 | 5.18 $\pm$.06 | 6.54 $\pm$.04 |
| Conv-Tasnet | ✓ | ✗ | 6.13 $\pm$.04 | 6.19 $\pm$.05 | 6.60 $\pm$.07 | 4.88 $\pm$.02 | 6.87 $\pm$.05 |
| Demucs | ✓ | 150 | 6.50 $\pm$.02 | 7.04 $\pm$.07 | 6.68 $\pm$.04 | 5.26 $\pm$.03 | 7.00 $\pm$.05 |
| Conv-Tasnet | ✓ | 150 | 6.57 $\pm$.02 | 7.35 $\pm$.05 | 6.96 $\pm$.08 | 4.76 $\pm$.05 | 7.20 $\pm$.05 |
| MMDenseLSTM | ✗ | 804 | 6.50 | 6.96 | 6.00 | 5.55 | 7.48 |

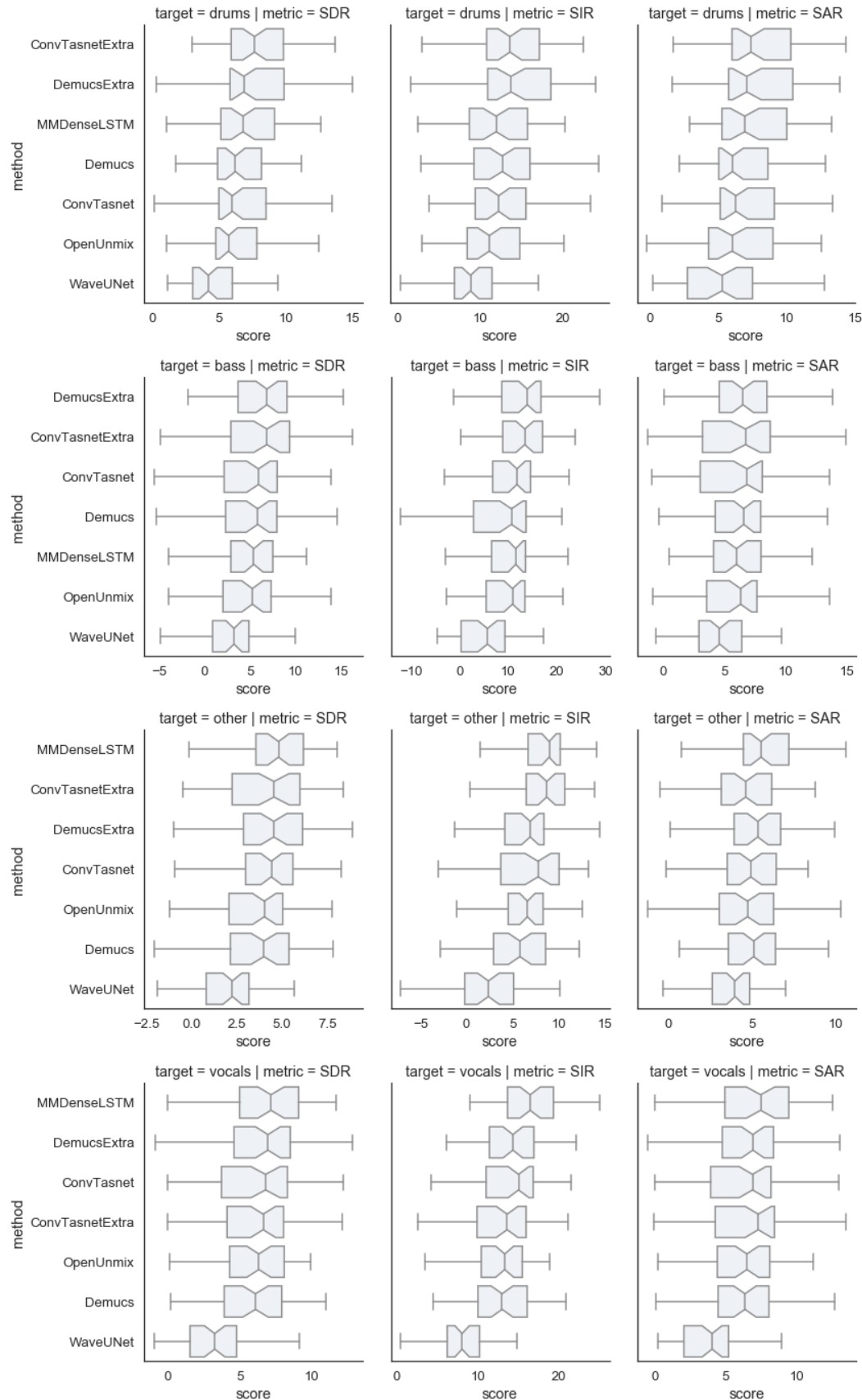

Figure 3: Boxplot showing the distribution of SDR, SIR and SAR over the tracks of the MusDB test.

