# OpenReview forum: "Music Source Separation in the Waveform Domain"
_ICLR.cc/2020/Conference — Reject_

### Official Review · AnonReviewer3 · 2019-10-24
**Official Blind Review #3**

**Rating:** 8

**Review:**

The authors present modifications to the state of the art (waveform based) source separation model (Wave-U-Net) and improve the state of the art to be comparable to spectral masking based methods. They clearly outline all the architectural changes they make (GLU nonlinearities, strided upsampling, bidirectional RNN), perform thorough evaluations against strong baselines, and a complete ablation study to demonstrate the value of each component. The paper reads well and quickly informs newcomers to the field with proper motivation and context. For all these reasons I think it should be accepted.  One weakness of the paper is that the relative incremental nature of the study, but given the thoroughness and clarity of the experiments, and the non-triviality of the architecture search, I think this is a valuable contribution for the ICLR community.

One small suggestion, the language difference between "upsampled" convolution and "transposed" convolution is a bit confusing. I might suggest focusing on the striding of the convolution vs. bilinear upsampling, as a non-strided convolution can still be "transposed" from a standard API point of view in pytorch or tensorflow.

**Experience Assessment:**

I have published in this field for several years.

**Review Assessment: Checking Correctness Of Derivations And Theory:**

N/A

**Review Assessment: Checking Correctness Of Experiments:**

I carefully checked the experiments.

**Review Assessment: Thoroughness In Paper Reading:**

I read the paper at least twice and used my best judgement in assessing the paper.

---

> ### Author Response · Authors · 2019-11-15
> **Reply to Reviewer 3**
>
> We thank the reviewer for the constructive and positive review.
>
> For the discussion of "upsamplied convolution" vs "transposed convolution", We clarify that what we mean by "upsample-convolution" is a (e.g., bilinear) upsampling method followed by a non-strrided convolution vs a strides .
>
> We also made significant updates of the paper (please see the general comments for a summary of the main changes).

---

### Official Review · AnonReviewer1 · 2019-10-24
**Official Blind Review #1**

**Rating:** 3

**Review:**

This paper suggests using a Unet type architecture to perform end-to-end source separation. They are reporting performance improvement over another architecture which uses Unets architecture in conjunction with Wavenet decoders. They also report a very marginal performance improvement over an STFT based model (open unmix) The improment is 0.02 dB (table 1), and I am not sure if it is statistically significant.


Also, there is no mention of algorithms which adaptively learn the basis and then do masking similar to what we do in the STFT domain. A very popular example for this is tasnet, which performs well on speech source separation tasks. I would like to see comparisons with this model. If you think there is a specific reason why not to use adaptive basis approaches such as TASNET, please do let me know.

**Experience Assessment:**

I have published in this field for several years.

**Review Assessment: Checking Correctness Of Derivations And Theory:**

N/A

**Review Assessment: Checking Correctness Of Experiments:**

I assessed the sensibility of the experiments.

**Review Assessment: Thoroughness In Paper Reading:**

I read the paper at least twice and used my best judgement in assessing the paper.

---

> ### Author Response · Authors · 2019-11-15
> **Reply to Reviewer 1**
>
> We thank the reviewer for pointing out the Tasnet paper. We made experiments with Tasnet and updated the paper accordingly. Please, see the general comments for more details.
>
> We also improved the model performance (mostly the inference procedure that takes an average over random initial shifts of the input). The results of Demucs (and Tasnet) are now very significantly better than the best models operating on spectrograms (+0.4 SDR points for Tasnet and +0.3 SDR for Demucs).

---

> > ### Comment · AnonReviewer1 · 2019-11-15
> > **Thanks for the honesty and the effort**
> >
> > Thanks for including the Tasnet result, and showing the results transparently. I am seeing that the models with your model are in the same ballpark as Tasnet, do we have any advantage with the proposed model compared to Tasnet? How do the computational requirements compare?

---

> > > ### Author Response · Authors · 2019-11-15
> > > **Comparison of Tasnet and Demucs**
> > >
> > > Computational requirements are of a similar order. Tasnet requires more memory and computation for the same duration but because it only uses local information it can be trained on much smaller audio extracts, which compensates in the end.
> > >
> > > Please have a listen to the audio samples included in the link. You will see that there is something like static noise on the output of Tasnet, as well as hollow attacks, especially noticeable on the drums and bass sources. We confirmed this observation with mean opinion scores evaluating both the quality (i.e. absence of artifacts) and purity (absence of contamination from other sources) of the separated audio. Tasnet has slightly better purity (3.42 against 3.30 for Demucs) but Demucs has significantly higher quality (3.22 against 2.85).
> > > Those results are detailed in Table 2 and 3 in the new version of the paper.

---

> > > > ### Comment · AnonReviewer1 · 2019-11-15
> > > > **SIR**
> > > >
> > > > I see, but I am also hearing more interference with Demucs, which might explain lower SDRs. Do you have numbers for SIR values?

---

> > > > > ### Author Response · Authors · 2019-11-15
> > > > > **SIR reply**
> > > > >
> > > > > There is significantly more interference with Demucs, you can find the SIR on page 15 of the paper, in the appendix. SIR for Demucs is 10.39, 11.47 for Tasnet.
> > > > > We don't pretend that Demucs is better than Tasnet, but we support that the two have different approaches and capabilities, one being better at separation and the other keeping better naturalness. While it would be ideal to have the best of both world, we believe it is beyond the scope of this paper. We provide a solid experimental comparison of spectrogram methods, Tasnet and Demucs, both with standard source separation metrics and human evaluations. We also provide the code for the training pipeline of Demucs and Tasnet on MusDB which we are going to open source.
> > > > > Overall, we agree with Reviewer 3 that this paper is not a revolution in the field but a solid foundation for future work by all interested parties.

---

### Official Review · AnonReviewer2 · 2019-10-27
**Official Blind Review #2**

**Rating:** 3

**Review:**

In this work, the authors consider the task of supervised music source separation, i.e., separating out the components (bass, drums, voice, other) out of a mixed music track. In particular, the work considers the task of supervised source separation where the individual target tracks are available during training time. The main contribution of this work is the improvement of an end-to-end waveform-to-waveform separation models through a number of architectural changes that allow such waveform-to-waveform models to perform comparably with other current state-of-the-art methods that instead operate in the spectrogram domain.

Overall, the paper is generally well written and the method is easy to follow. However, I have a couple of concerns about this work, based on which I would rate the work as “weak reject”:

1. The specific architecture proposed in this task does appear to improve performance for this particular task, but it is not clear to me that the conclusions drawn based on the study in this work will be generally applicable to other related tasks. Also, the final model is still only comparable to the spectrogram-based methods overall, and appears to do a slightly worse job in separating the ‘vocal’ and ‘other’ tracks than these baseline methods. As such, I’m unsure about the impact of this work.
2. The section describing the evaluation metric SDR wasn’t very clear to me. Perhaps it would be better to just refer back to (Vincent et al., 06) and just describe what SDR captures instead? Or alternatively, more details could be added to explain the computation more clearly.
3. Section 4.2: The authors mention that they multiply each source by +/- 1. I didn’t follow why this is being done. Could the authors please clarify.

Minor comments:
1. The notation in Equation 1 is slightly confusing because ‘s’ is used to index both the mixture and the sources. I think x \in \mathcal{D} would be clearer. Similarly I wonder if x_s \in \mathbb{R}^C \times \mathbb{R}^T would be clearer than x_s \in \mathbb{R}^{C,T}
2. The reference (Oord et al., 2017) should be rendered as (van den Oord et al., 2017)
3. The Figure uses K, and S to denote Kernel Width and stride, but this isn’t explicitly mentioned in the text. Perhaps it would be useful, in Section 3.1 to write: “... convolution with kernel width, K=8, and stride, S=4, ...” or something similar, and updating the caption to explain the notation.
4. Section 3.2: “The decoder is almost the symmetric of the encoder” --> “The decoder is almost the inverse of the encoder” would perhaps be better?
5. Section 3.2: “The final layer … S.C_0 ...” --> “The final layer … S * C_0 ...”
6. The notation in Equation 1 and Equation 2 is slightly inconsistent (lower-case vs. upper-case L for the loss function)
7. Section 4.3: “additionnaly” --> “additionally”; “32 GB or RAM” --> “32 GB of RAM”


**Experience Assessment:**

I have read many papers in this area.

**Review Assessment: Checking Correctness Of Derivations And Theory:**

I assessed the sensibility of the derivations and theory.

**Review Assessment: Checking Correctness Of Experiments:**

I assessed the sensibility of the experiments.

**Review Assessment: Thoroughness In Paper Reading:**

I read the paper thoroughly.

---

> ### Author Response · Authors · 2019-11-15
> **Reply to reviewer 2**
>
> We thank the reviewer for the constructive comments. The reviewer's main points were:
>
> 1- "it is not clear to me that the conclusions drawn based on the study in this work will be generally applicable to other related tasks"
>
> We argue that there are two contributions in that paper:
> a) showing that now the state-of-the-art is on multi-instrument source separation is obtained by wave-form-to-waveform models (see general comments for updated results)
> b) showing that approaches based on convolutional encoders/decoders are effective for source separation tasks. Of course, experiments on other source sepraration problems with similar architectures would be a big plus, but it is out of the scope of the current paper and we leave such studies as future work.
>
> 2-"The section describing the evaluation metric SDR wasn’t very clear to me"
>
> We rephrased it and followed the reviewer suggestion to refer to Vincent et al. and only explain what the metric is supposed to capture
>
> 3- multiplying by +1 or -1 corresponds to a global shift of phase. we use that as a form of data augmentation, since the model should be equivalent to such a shift
>
> We also updated the paper to correct the reviewer's minor comments.

---

### Author Response · Authors · 2019-11-15
**general comment: comparison to Tasnet and update of the paper**

We first would like to thank the reviewers for their constructive comments. We made significant updates to the paper to respond to the reviews:

1- We performed several improvements to the model, and updated the corresponding empirical study (increased the number of channels, changed the batch size, and updated the inference procedure using an average over random initial shifts of the song (see Section 4.5). The results are now of 5.6 SDR, while the best previous results on MusDB are 5.3 (see the updated Table 1). You might notice that the validation loss has increased between the two versions, this is because we initially used pytorch DistributedSampler which repeats some of the tracks to make the dataset a multiple of the number of GPUs. This biased the validation loss in the first version of the paper and we fixed that since.

2- We added experiments with Tasnet, as suggested by reviewer 1. Because Tasnet was developed for monophonic source separation with a sampling rate of 8kHz, we had to adapt it to make it work on stereo sampled at 44kHz (which requires increasing the receptive field of the overall network to obtain reasonable performances). With these modifications (described in a new Section 3), Tasnet obtains state-of-the-art SDR of 5.7, slightly better than Demucs. The difference diminishes when adding more training data, as they both obtain 6.3 SDR with 150 additional songs, whereas the previously best reported result on MusDB test set was 6 with 800 additional songs.

3- We performed additional human evaluations between Tasnet and Demucs (trained without additional data) in terms of quality and contamination with other sources. The lower SDR for Demucs is explained by a bit more contamination. On the other hand, Demucs obtains significantly better results in terms of quality.

4- We updated the paper overall to reorganize the discussion on the differences between Demucs and Tasnet instead of the differences between Demucs and Wave-U-Net.

5 - We added the source code to the supplementary material. Pre-trained models are also available for evaluation and can be downloaded on demand from the demucs.separate script. All instructions the README file in the code folder in the ICLR code link. The link will download both the code and audio samples for Demucs, Conv-Tasnet and all the baselines. The entire supplementary material is 100MB and might take a few minutes to download.

---

### Decision · Program_Chairs · 2019-12-19

**Decision:**

Reject

**Comment:**

The paper proposed a waveform-to-waveform music source separation system. Experimental justification shows the proposed model achieved the best SDR among all the existing waveform-to-waveform models, and obtained similar performance to spectrogram based ones. The paper is clearly written and the experimental evaluation and ablation study are thorough. But the main concern is the limited novelty, it is an improvement over the existing Wave-U-Net, it added some changes to the existing model architecture for better modeling the waveform data and compared masking vs. synthesis for music source separation.